# Interferometric control of magnon-induced nearly perfect absorption in cavity magnonics

J. W. Rao [1], P. C. Xu[1], Y. S. Gui[1], Y. P. Wang [1], Y. Yang[1], Bimu Yao [2,3]✉, J. Dietrich[4], G. E. Bridges[4], X. L. Fan[5], D. S. Xue[5] & C.-M. Hu [1]✉

The perfect absorption of electromagnetic waves has promoted many applications, including photovoltaics, radar cloaking, and molecular detection. Unlike conventional methods of critical coupling that require asymmetric boundaries or coherent perfect absorption that require multiple coherent incident beams, here we demonstrate single-beam perfect absorption in an on-chip cavity magnonic device without breaking its boundary symmetry. By exploiting magnon-mediated interference between two internal channels, both reflection and transmission of our device can be suppressed to zero, resulting in magnon-induced nearly perfect absorption (MIPA). Such interference can be tuned by the strength and direction of an external magnetic field, thus showing versatile controllability. Furthermore, the same multi-channel interference responsible for MIPA also produces level attraction (LA)-like hybridization between a cavity magnon polariton mode and a cavity photon mode, demonstrating that LA-like hybridization can be surprisingly realized in a coherently coupled system.

[1] Department of Physics and Astronomy, University of Manitoba, Winnipeg, Canada R3T 2N2. [2] State Key Laboratory of Infrared Physics, Shanghai Institute of Technical Physics, Chinese Academy of Sciences, Shanghai 200083, China. [3] School of Physical Science and Technology, ShanghaiTech University, Shanghai 201210, China. [4] Department of Electrical Engineering, University of Manitoba, Winnipeg, Canada R3T 2N2. [5] The Key Lab for Magnetism and Magnetic Materials of Ministry of Education, Lanzhou University, Lanzhou 730000, China. ✉email: yaobimu@mail.sitp.ac.cn; hu@physics.umanitoba.ca

In optoelectronic devices and microwave systems, perfect electromagnetic absorption via light-matter interactions is desirable so that energy conversion or information transfer can be maximized. Successful examples in practical applications include photon detectors/imagers[1–6] and stealth technology[7]. However, the single-beam absorption of a resonant mode with a symmetric boundary never goes beyond 50%[8–11] (Fig. 1a, details in supplementary Note 1). This reflects a common limitation shared by planar plasmas[8], qubits[9], graphene[10], and thin films[11], which can be generally abstracted into a symmetric two-port system. There are two methods of external operation that can overcome this limitation. The first is to break the boundary symmetry to form a single-port reflector. Once the energy-feeding rate equals the device loss (critical coupling)[12,13], destructive self-interference can eliminate the reflection and lead to perfect absorption (Fig. 1b). Another method is called coherent perfect absorption (CPA) or time-reversed lasing[14–19]. This can keep the system's boundary unchanged, enabling the linear control of light with light[15], whereas the waveform manipulation of multiple coherent incident beams is essential (Fig. 1c).

These two methods possess their exclusive advantages, either the simplicity of single-beam input or the adjustability of multiple ports. To obtain their complementary advantages, multiple internal channels of a system, instead of multiple incident beams, can be used. The challenge stems from the complexity of constructing internal mode structures. Thus, this method has rarely been explored experimentally. Emerging cavity magnonics[20–29] constitutes an attractive platform to coherently tune hybrid photon-magnon states. In a cavity magnonic system, microwave photons couple to magnons (the quanta of collective spin excitations) and evolve into quasi-particles called cavity magnon polaritons (CMPs)[23], whose hybridization, dispersion, and damping rates can be flexibly tailored by harnessing the magnon dynamics in solids. Utilizing the dynamics of CMPs, applications have flourished, such as gradient memories[30], polaritonic logic gates[31], spin current manipulation[32], and information processing at the quantum limit[33]. In particular, with two input beams as effective gains, CPA has been realized in a coupled magnon-photon system[34]. This motivates us to further explore solutions for single-beam perfect absorption in cavity magnonics.

In this work, we demonstrate that single-beam nearly perfect absorption can be achieved within a multi-port cavity magnonic device by exploiting the interference between its tunable internal channels (polariton modes). A cross cavity with distinct dipole (D-) and quadrupole (Q-) modes is designed. By coupling it to a magnon mode (M), the interference between D- and Q-modes is mediated by the magnon mode. This interference is tunable by both the strength and the direction of an external magnetic field, and can modify magnetically induced transparency (MIT) into a level attraction (LA)-like hybridization in the transmission. Different from previously reported systems with dissipative coupling[35–46], negative mass[47], or negative energy[48–53], LA-like hybridization in our device is a special transmission response produced by destructive interference. This hybridization eliminates transmission and allows perfect absorption (96% in the experiment and 100% in the theory reported in this work). We call this phenomenon magnon-induced nearly perfect absorption (MIPA). Our work stimulates the idea of interferometric control in multi-channel dynamics, and may bring insights into perfect absorption in hybrid systems. The techniques developed in our work may find applications in microwave photon traps, detectors, or transducers.

## Results

**Constructing the MIPA architecture.** We design a planar cross cavity consisting of two orthogonal coplanar waveguide resonators, as sketched in Fig. 1d. The excitation of each coplanar waveguide resonator produces two degenerate D-modes[31]. The fundamental excitation of the whole structure produces the Q-mode (see supplementary Note 2 for details). Our cross cavity is a prototypical two-port device. The scattering (S-) parameters measured by Vector Network Analyzer (VNA) with an excitation power of −5 dBm from port-1 and port-2 are shown in Fig. 1e and f with two other ports shorted. The D- and Q-modes occur at $\omega_d/2\pi = 3.475$ GHz and $\omega_q/2\pi = 3.675$ GHz, respectively. Their radio frequency (rf) magnetic field ($h$) distributions are calculated by numerical simulations and presented in two insets. These two cavity modes provide two resonant channels for the input microwave. They are in phase in the reflection but out of phase in the transmission, according to Fig. 1f. The resulting opposite

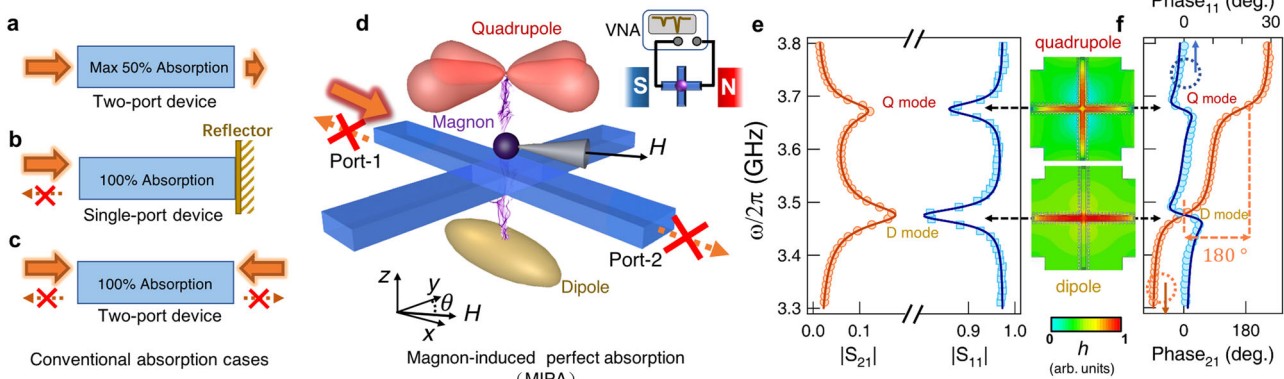

**Fig. 1 A sketch of MIPA. a** Schematic diagram of the electromagnetic absorption limitation (50%) in a symmetric two-port system. **b** Perfect electromagnetic absorption of a single-port system under the critical coupling condition. **c** Schematic diagram of the coherent perfect absorption (CPA) strategy that utilizes the interference of two coherent counter-propagating beams. **d** A schematic of MIPA. Two cavity photon modes in a cross cavity are coupled to a magnon mode in a YIG sphere. The bias magnetic field $H$ is applied in the x-y plane with an adjustable angle $\theta$ from y-axis. By utilizing the interference between hybrid photon-magnon modes, near-perfect absorption can be achieved. **e** Transmission and reflection spectra of the cross cavity show two distinct cavity photon modes, i.e., D- and Q-modes (labeled near their resonances). Symbols and solid lines show experimental and calculated results, respectively. **f** Phase information of these two modes. They are out of phase in the transmission but in phase in the reflection. The two insets between (**e**) and (**f**) are numerical simulations of the radio frequency magnetic field ($h$) distributions for D- and Q-modes.

interference behavior in two scattering directions is the key to realizing MIPA in our device.

Although the D- and Q-modes are detuned about $(\omega_q - \omega_d)/2\pi = 200$ MHz, the interference between them can be mediated by a magnon mode. In the experiment, an yttrium iron garnet (YIG, $Y_3Fe_5O_{12}$) sphere with a diameter of 1 mm is placed at the intersection of the cross cavity. A bias magnetic field ($H$) with a tunable angle ($\theta$ from the y-axis shown in Fig. 1d) is applied in the x-y plane to control the excitation of the magnon mode ($\omega_m$). This coupled cavity photon-magnon system can be modeled by the following Hamiltonian (see details in Supplementary Note 3):

$$
\begin{aligned}
H_{sys} = {} & \hbar\omega_d \hat{d}_x^\dagger \hat{d}_x + \hbar\omega_d \hat{d}_y^\dagger \hat{d}_y + \hbar\omega_m \hat{m}^\dagger \hat{m} + \hbar\omega_q \hat{q}^\dagger \hat{q} \\
& + \hbar g_d \sin\theta(\hat{d}_x^\dagger \hat{m} + \hat{d}_x \hat{m}^\dagger) + \hbar g_d \cos\theta(\hat{d}_y^\dagger \hat{m} + \hat{d}_y \hat{m}^\dagger) \\
& + \hbar g_q(\hat{q}^\dagger \hat{m} + \hat{q}\hat{m}^\dagger)
\end{aligned}
\tag{1}
$$

where $\hat{d}_{x,y}$ ($\hat{d}_{x,y}^\dagger$) represents the annihilation (creation) operators of degenerate D-modes along the $x, y$-directions, i.e., the horizontal or perpendicular D-modes. In addition, $\hat{q}$ ($\hat{q}^\dagger$) and $\hat{m}$ ($\hat{m}^\dagger$) represent the annihilation (creation) operators of the Q-mode and the magnon mode, respectively. Because the $rf$-magnetic fields of the two degenerate D-modes are linearly polarized and orthogonal (shown in Supplementary Note 2), coupling strengths between them and the magnon mode are determined by the projection of the $rf$-field onto the spin precession plane[32], given by $g_d \sin\theta$ and $g_d \cos\theta$. By contrast, the coupling between the Q- and magnon modes is independent of the magnetic field direction. Therefore, it can be represented by a constant $g_q$.

Considering each mode's damping rate and their coupling effects with external photon baths[54], the coupled dynamics derived from the Hamiltonian follow

$$
\frac{d}{dt}
\begin{bmatrix}
\hat{d}_x \\ \hat{d}_y \\ \hat{m} \\ \hat{q}
\end{bmatrix}
= -i\Omega
\begin{bmatrix}
\hat{d}_x \\ \hat{d}_y \\ \hat{m} \\ \hat{q}
\end{bmatrix}
+
\begin{bmatrix}
\sqrt{\kappa_d} \\ 0 \\ 0 \\ \sqrt{\kappa_q}
\end{bmatrix}
p_1^{in}
\tag{2}
$$

with

$$
\Omega =
\begin{bmatrix}
\widetilde{\omega}_d & 0 & g_d\sin\theta & 0 \\
0 & \widetilde{\omega}_d & g_d\cos\theta & 0 \\
g_d\sin\theta & g_d\cos\theta & \widetilde{\omega}_m & g_q \\
0 & 0 & g_q & \widetilde{\omega}_q
\end{bmatrix}
\tag{3}
$$

where $\widetilde{\omega}_d = \omega_d - i(\kappa_d + \gamma_d)$, $\widetilde{\omega}_m = \omega_m - i\gamma_m$ and $\widetilde{\omega}_q = \omega_q - i(\kappa_q + \gamma_q)$ are the complex frequencies of the uncoupled D-, magnon, and Q-modes. $\kappa_{d,q}$ and $\gamma_{d,q}$ respectively represent the external and intrinsic damping rates of the D-modes and the Q-mode. $\gamma_m$ is the intrinsic damping rate of the magnon mode. $p_1^{in}$ is the input signal from port-1. Based on the characterized phase difference between the D-mode and the Q-mode (Fig. 1 (d)), the reflection detected at port-1 can be derived from $S_{11} = 1 - (\sqrt{\kappa_d}\hat{d}_x + \sqrt{\kappa_q}\hat{q})/p_1^{in}$, while the transmission detected at port-2 can be derived from $S_{21} = (\sqrt{\kappa_d}\hat{d}_x - \sqrt{\kappa_q}\hat{q})/p_1^{in}$. With $\omega_m = 0$, the calculated reflection and transmission spectra of the empty cavity are plotted as solid lines in Fig. 1 (e) and (f), which well reproduce our data with $\kappa_d/2\pi = 3.3$ MHz, $\kappa_q/2\pi = 1.9$ MHz and $\gamma_d/2\pi = \gamma_q/2\pi = 14.5$ MHz.

Obviously, as long as $\sqrt{\kappa_d}\hat{d}_x = \sqrt{\kappa_q}\hat{q} = p_1^{in}/2$, perfect absorption is achieved. In this condition, both reflection ($S_{11}$) and

transmission ($S_{21}$) vanish, so that the input signal can be completely dissipated in the cavity, i.e., the absorption ($A$) is unity, $A = 1 - |S_{11}|^2 - |S_{21}|^2 = 1$. To meet this condition, one may consider adjusting both the degeneracy and intensities of the two channels by redesigning the cavity structure, similar to the idea of superscattering in metamaterials[55,56]. Here we demonstrate an easier method based on a cavity magnonic approach[34,57,58], utilizing a tunable magnon mode to precisely control electromagnetic absorption by tuning either the strength or direction of the external magnetic field.

**Interference-induced LA-like hybridization.** To better understand the interference in our multi-channel device, we choose the transmission spectra as an example and study the magnon-mediated interference in two steps. First, we focus on the strong coupling between two D-modes and the magnon mode near $\omega_d$ (the D–M coupling). Because $g_q \ll g_d$ and the Q-mode is far detuned from the D-modes, the influence of the Q-mode on the D–M coupling is negligible. Three CMP modes produced by D–M coupling can be analytically solved from Eq. (2) as $\omega_d$ and $\omega_\pm = \text{Re}[\widetilde{\omega}_d + \widetilde{\omega}_m \pm \sqrt{(\widetilde{\omega}_d - \widetilde{\omega}_m)^2 + 4g_d^2}]/2$ (as shown in Fig. 2a, more details in Supplementary Note 4). The frequencies of the three CMP modes are independent of the direction of the external magnetic field, i.e., $\theta$, but their mode intensities are tunable by $\theta$ following the relations of $I_c \propto \cos^2\theta$ and $I_\pm \propto \sin^2\theta$, where $I_c$ and $I_\pm$ correspond to the central and side mode intensities (derivations shown in supplementary Note 4). When the external magnetic field is pointed in the $y$-direction ($\theta = 0°$), $g_d \sin\theta$ equals zero. The magnon mode is decoupled to the horizontal D-mode ($\hat{d}_x$). The upper- and lower-branch CMP modes ($\omega_\pm$) become dark ($I_\pm \approx 0$), while the central CMP mode at $\omega_d$ is bright, represented by black dashed lines in Fig. 2b. Then we rotate the external magnetic field to the $x$-direction ($\theta = 90°$). $g_d \sin\theta$ reaches its maximum (i.e., $2\pi \times 65.6$ MHz). The magnon mode is strongly coupled to the horizontal D-mode ($\hat{d}_x$), which is driven by the input signal $p_1^{in}$. In this case, the central hybrid mode at $\omega_d$ vanishes, but both upper- and lower-CMP modes ($\omega_\pm$) become bright (Fig. 2c), which are ready for magnon mediation in subsequent operations.

After characterizing the D–M coupling, we move on to the second step, i.e., the coupling between the Q- and magnon modes (the Q–M coupling) in the vicinity of $\omega_q$. The influence of strong D–M coupling must be considered in Q–M coupling. Therefore, the Q–M coupling can be treated as the coupling between the upper-CMP mode ($\omega_+$) and the Q-mode (Fig. 2a). Switching on/off the intensity of the upper-CMP mode ($I_+$) can be utilized here to manipulate the Q–M coupling. To describe this behavior, we reduce Eq. (2) to a more concise form with only the upper-CMP mode and the Q-mode (see the Supplementary Note 6 for more details).

$$
\begin{bmatrix}
\omega - \widetilde{\omega}_+ & -g_q \\
-g_q & \omega - \widetilde{\omega}_q
\end{bmatrix}
\begin{bmatrix}
\hat{m} \\ \hat{q}
\end{bmatrix}
= i
\begin{bmatrix}
\sqrt{\kappa_+} \\ \sqrt{\kappa_q}
\end{bmatrix}
p_1^{in}
\tag{4}
$$

where $\kappa_+ = \frac{g_d^2 \sin^2\theta}{(\omega - \widetilde{\omega}_-)^2}\kappa_d$ is the effective coupling strength between the upper-CMP mode and the extra-cavity photon bath. This strength originates from the D–M coupling, and shows a strong $\theta$ dependence. At $\theta = 0°$, $\kappa_+ = 0$, so that the channel for the input signal ($p_1^{in}$) sustained by the D–M coupling is shut down. The Q-M coupling becomes a conventional coherent coupling case ($g_q/2\pi = 4$ MHz) with a single input channel via the Q-mode. From transmission spectra, a magnetic induced transparency (MIT)-like behavior is observed (Fig. 2d) because of the criterion of

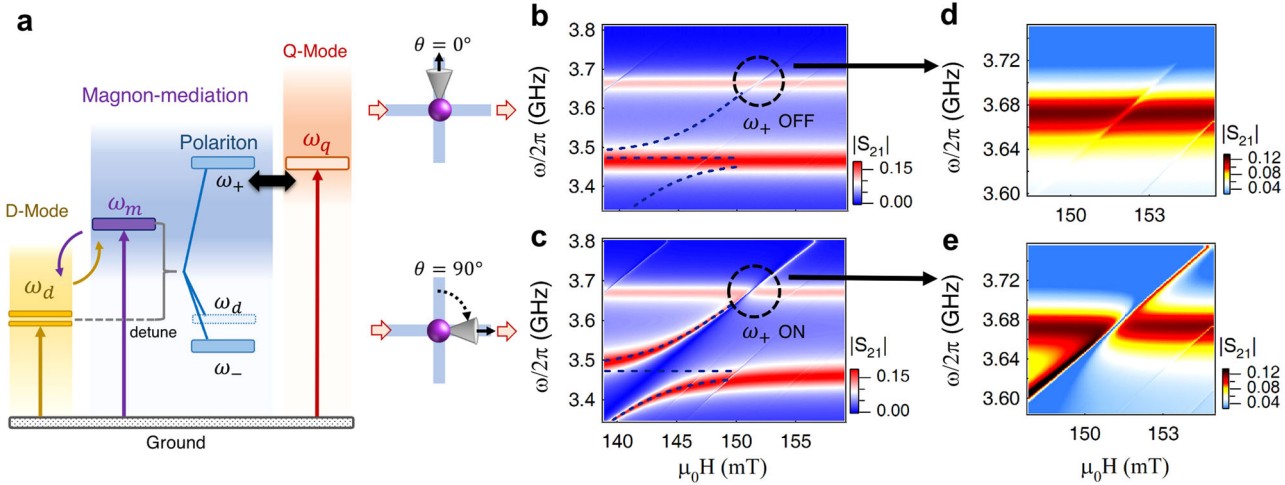

**Fig. 2 Interference between internal photon-magnon hybrid modes. a** Schematic picture of the energy level diagram in our MIPA system. **b, c** shows the switching off/on of the interference between the upper-CMP mode ($\omega_+$) and the Q-mode ($\omega_q$) in transmission by tuning $\theta$ from 0° to 90°. **d** Conventional MIT-like behavior occurs when the channel sustained by the upper-CMP mode ($\omega_+$) is off. **e** LA-like hybridization occurs when the channel sustained by the $\omega_+$ mode is on.

$\gamma_+ < g_q < \gamma_q + \kappa_q$, where $\gamma_+$ is the intrinsic damping rate of the upper-CMP mode. As we tune $\theta$, $\kappa_+$ increases in proportion to $\sin^2\theta$. The channel sustained by the D–M coupling is opened up. Consequently, in addition to the coherent coupling effect between the upper-CMP mode and the Q-mode, interference between two resonant channels emerges. In the transmission, these two effects lead to the coalescence of two resonant peaks (red color) and produce LA-like hybridization (Fig. 2e). In addition to two resonant peaks, a sharp dip (blue color) occurs between two peaks, where the transmission is suppressed to zero. By utilizing it, MIPA can be achieved, which will be discussed in detail later. Using Eq. (4) and the input-output relation of $S_{21} = (\sqrt{\kappa_+}\hat{m} - \sqrt{\kappa_q}\hat{q})/p_1^{in}$, transmission mappings in both cases (Fig. 2d, e) can be well reproduced (see Supplementary Note 6).

Intuitively, LA in our device can be understood by using a coherently coupled pendulum system with two driving forces. Due to the destructive interference induced by double drives, the maximum response of the system deviates from its eigenfrequencies in the frequency domain, and thus LA-like hybridization occurs (see Supplementary Note 5). This unique hybridization behavior in cavity magnonics was previously observed in 1D Fabry-Perot cavities[37,45], cavity anti-resonance[40], planar cavities[41–43], and two-tone driven CMPs[38,44]. In these works, LA was treated as a signature of the dissipative coupling effect. Although still rooted in the dissipative process, in our work, this unique hybridization behavior is achieved through destructive interference between internal channels sustained by polariton modes.

**Magnon-induced nearly perfect absorption.** By utilizing the interference in our device, MIPA can be realized. In addition to the suppression in the transmission, the reflection also needs to be suppressed to zero so that all input power can be absorbed and then dissipated in our device. At $\theta = 0°$, the interference between the two internal channels is off. The measured squared amplitudes of the reflection and transmission at the zero detuning ($\Delta = \omega_+ - \omega_q = 0$) are plotted in Fig. 3a, b. As a normal case of coherent coupling, an enhancement occurs in the reflection (a small peak in the frequency range marked in gray), accompanied by suppression in the transmission (a small dip). This opposite behavior in two scattering directions makes perfect absorption impossible. Figure 3c shows the measured absorption spectra at

different bias fields, from which no evident enhancement of the absorption is observed.

With $\theta$ increasing from 0° to 90°, interference mediated by the magnon mode gradually occurs. This interference effect nearly doubles the absorption rate, as shown by the absorption spectra measured at zero detuning with increasing $\theta$ (Fig. 3d). Typically, at 90°, the enhanced absorption comes from the fact that both squared amplitudes of the reflection and transmission are suppressed (shown in Fig. 3e, f, respectively, with the corresponding frequency range indicated by gray color). This shows an evident difference from the aforementioned MIT behavior. Consequently, an enhancement of microwave absorption occurs, as shown by the red spot in Fig. 3g.

Generally, the demonstration of absorption enhancement embodies magnon-mediated interference between two internal channels. However, further optimization is still needed to achieve perfect absorption. Based on the aforementioned condition of $\sqrt{\kappa_d}\hat{d}_x = \sqrt{\kappa_q}\hat{q} = p_1^{in}/2$ and Eq. (4), a perfect absorption condition is achieved as $\kappa_+\kappa_q \approx g_q^2 + \gamma_+\gamma_q - \sigma\Delta$, where $\sigma$ represents the small frequency difference between MIPA and $\omega_+$. Quantitatively, for the currently used cross cavity, the left term $\kappa_+\kappa_q \approx 6 \times 4\pi^2$ MHz$^2$ is much smaller than the right term ($40 \times 4\pi^2$ MHz$^2$). To realize perfect absorption, larger $\kappa_+$ and $\kappa_q$ are necessary. Considering the linear relation between $\kappa_+$ and $\kappa_d$, this problem can be solved by adjusting the coupling gaps of cavity ports.

Figure 4a shows a theoretical calculation of the absorption rate at zero detuning ($\Delta = 0$) by varying $\kappa_d$ and $\kappa_q$. We notice that the absorption rate can be enhanced to 100% with a slight increase in $\kappa_d$ and $\kappa_q$. The white dashed line in Fig. 4a indicates the region in which the absorption rate is over 90%. Nearly perfect absorption can be achieved throughout a wide range of these two parameters, which offers a large fabrication tolerance for cavity fabrication. Accordingly, we make another cross cavity, keeping the same design but slightly increasing $\kappa_d$ and $\kappa_q$. At the condition of $\theta = 90°$, as the upper-CMP mode $\omega_+$ approaches the Q-mode, drastic suppression of both the squared amplitudes of the reflection and transmission occurs, as shown in Fig. 4b. Therefore, an optimized absorption up to 96% is achieved (Fig. 4c), which unambiguously exhibits the possibility of realizing MIPA. Based on our theoretical model, all spectra can be well reproduced, shown as solid lines in Fig. 4b, c (see details in Supplementary Note 7). We

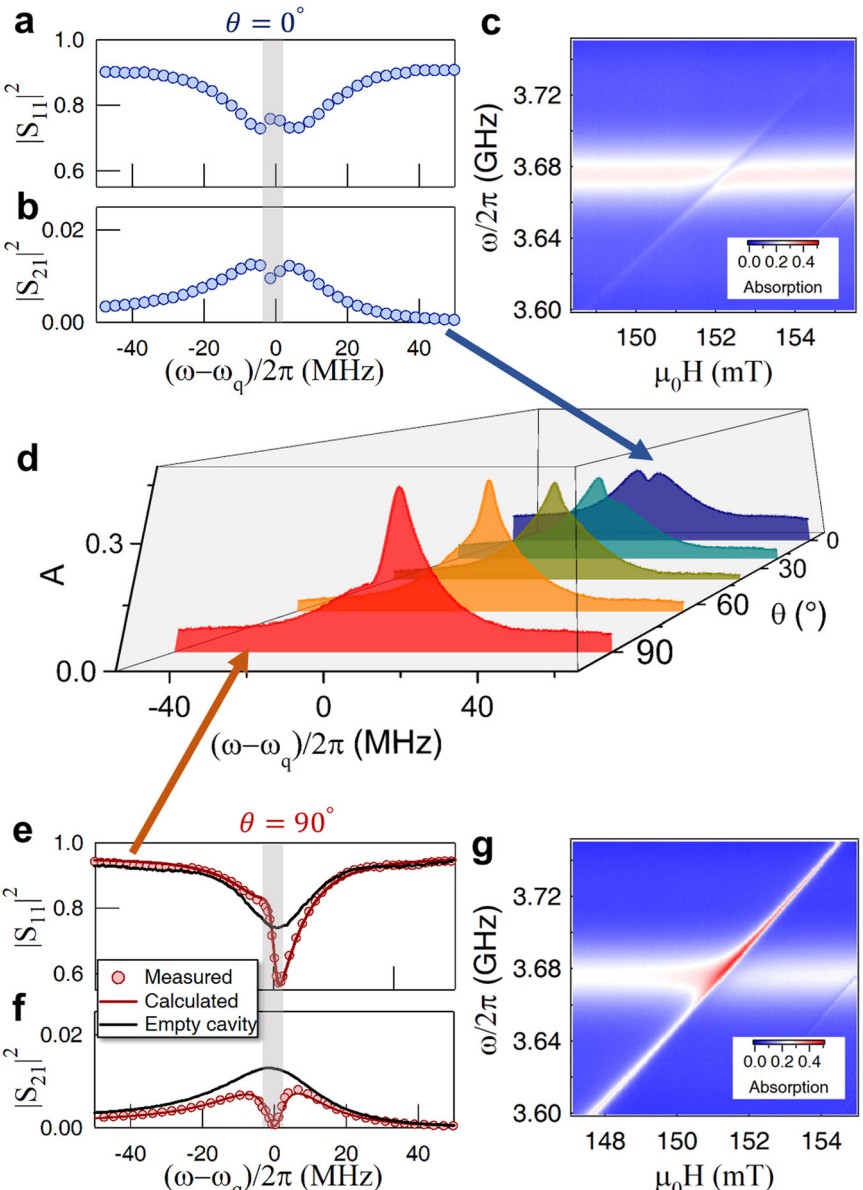

**Fig. 3 Absorption enhancement.** At $\theta = 0°$, corresponding to conventional MIT dynamics, the squared amplitudes of the reflection ($|S_{11}|^2$) enhancement and transmission ($|S_{21}|^2$) suppression at zero detuning are plotted in (**a**) and (**b**). Measured absorption spectra at different bias fields are plotted in (**c**). By tuning $\theta$, the measured absorption spectra at 5 different angles are plotted in (**d**). Typically, at $\theta = 90°$, both squared amplitudes of the reflection (**e**) and transmission (**f**) are suppressed relative to the empty cavity (black solid lines). Red solid lines in (**e**) and (**f**) indicate the calculated results. Absorption spectra measured at different bias magnetic fields are plotted in (**g**), which shows an evident absorption enhancement (the red spot).

note that, since our model takes advantage of the linear harmonic dynamics, this theory is promising to be generalized to other harmonic systems, or scaled down to the quantum regime.

In conclusion, we realize near-perfect absorption by utilizing magnon-mediated two-channel interference in a cavity magnonic device, i.e., MIPA. Unlike conventional CPA techniques that manipulate the interference of multiple coherent input signals, MIPA in our device needs only a single input signal. Due to the high tunability of the magnon mode, our method exhibits flexibility in controlling the absorption by using an external magnetic field. By simultaneously suppressing our device's reflection and transmission, nearly perfect absorption (96%, close to the predicted value of 100%) is achieved. By implementing the spin pumping effect[26] in the magnetic material, MIPA could allow the incoming microwave to be maximally utilized for the generation and manipulation of spin currents which can be

further electrically detected for the magnon-based energy harvesting and sensing. The extrinsic damping rates of the upper-branch polariton mode $\kappa_+$ and the Q-mode $\kappa_q$, may also be used as effective gains to construct a parity-time-symmetric system[34]. This idea will be useful for designing tunable modulators and switches with knowledge enriched by pseudo-Hermiticity and exceptional-point physics. Additionally, because of the destructive interference in transmission, an LA-like hybridization between a CMP mode and a cavity photon mode is observed, which offers a versatile method for the realization of LA in coherent information processing.

## Methods
**Device description.** The cross cavity used in our experiment is fabricated on a 0.762 mm thick RO4350B substrate, consisting of two half-wavelength coplanar waveguide resonators with a length of 25 mm and a central conductor width of

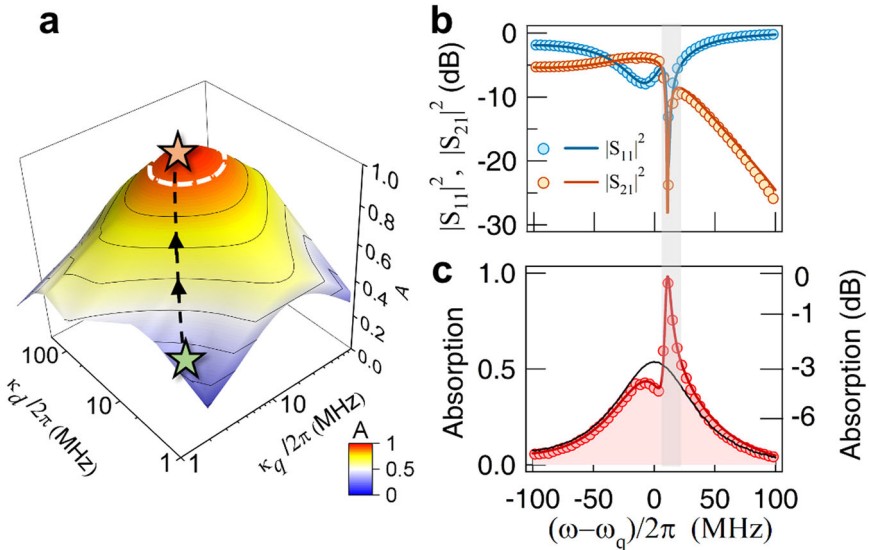

**Fig. 4 Near-perfect absorption. a** Theoretical calculation of the maximal absorption of our cavity magnonic device as a function of $\kappa_d$ and $\kappa_q$. The white dashed line indicates the region in which the maximal absorption rate is over 90%. Green and yellow sparks indicate the parameters of the currently used and modified cross cavities, respectively. **b** Measured square of reflection ($|S_{11}|^2$ in blue circles) and transmission ($|S_{21}|^2$ in orange circles) of the modified cavity magnonic device, shown in dB scale. MIPA occurs when two dips coalesce. Correspondingly, the measured absorption spectrum is plotted in **c** in red circles, which shows enhancement compared with empty cavity absorption (black solid line). The measured maximal absorption rate is 96%. Solid orange and blue lines in **b** and red lines in **c** are results calculated by using Eq. (2).

1.14 mm. Input/output ports are coupled to the cross cavity via coupling gaps. For the first cross cavity, the coupling gaps' width is 0.2 mm. $\kappa_d$ and $\kappa_q$ are designed to be small to maintain high quality (Q-) factors of the D- and Q-modes. Using this device, both the mode hybridization and interference behaviors can be clearly presented. For the modified cross cavity, the width of coupling gaps is designed as 50 $\mu$m to enhance the $\kappa_d$ and $\kappa_q$. This cavity is designed to demonstrate MIPA, so that Q-factors of cavity modes are not the main concern. The magnon mode used in our experiment is the Kittel mode supported by a 1 mm diameter YIG sphere. It follows the linear dispersion $\omega_m = \gamma(H + H_A)$, where $\gamma$, $H_A$, and $H$ represent the gyromagnetic ratio, anisotropy field, and static bias magnetic field, respectively. During the experiment, our cavity magnonic device is placed on a rotatable platform, so that the direction of the bias magnetic field relative to our device can be precisely controlled.

**Theory**. Details of theoretical derivations of MIPA are provided in Supplementary Notes 1–4.

## Data availability
The data that support the findings of this study are available from corresponding authors upon reasonable request.

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

## Acknowledgements

This work has been funded by NSERC Discovery Grants and NSERC Discovery Accelerator Supplements (C.-M.H.). J.W.R. is supported by CSC scholarship. B.M.Y. is supported by the National Natural Science Foundation of China under Grant Nos.11804352 and 11974369, the Youth Innovation Promotion Association CAS (2020247), SITP Innovation Foundation (CX-350), Strategic Priority Research Program of CAS (No. XDB43010200) and Shanghai Science and Technology Committee (No. 18JC1420401). We would like to thank Michael Harder and Yutong Zhao for discussions, and also acknowledge CMC Microsystems for providing equipment that facilitated this research. B.M.Y. thanks, in particular, the support and company from Dr. Y.L. Zhang.

## Author contributions

J.W.R., P.C.X., Y.P.W., and Y.Y. prepared the samples, performed the measurements and data analysis. J.W.R., Y.S.G., B.M.Y., and C.M.H. worked out the theory. J.W.R., Y.S.G., B.M.Y., and C.M.H. contributed to the writing of the paper. J.D., G.E.B., X.L.F., and D.S.X. provided support and conceptual advice. B.M.Y. and C.M.H. supervised this work.

## Competing interests

The authors declare no competing interests.
