## [Peer Review File · Nature Communications]

Reviewers' Comments:

Reviewer #1:

Remarks to the Author:

The manuscript presents a method for achieving magnon-induced perfect absorption (MIPA) using a cross cavity and a YIG sphere. The MIPA is achieved through tuning the interference between the hybridized modes which occur via coupling between the YIG magnon mode with the dipolar and quadrupolar photon modes of the cross cavity. The results are demonstrated well through the achievement of 96% power absorption, and the theoretical analysis of the hybridized system seems to model the data in a sufficient manner. Overall, the work presents a novel advancement in the field of cavity magnonics, and is theoretically sound. However, I have reservation about the conclusions drawn from their experimental observations, and would like to see more consistency within the data and more details on the data analysis before recommending publication in Nature Communications.

There are some issues with the manuscript:

- The maximum absorption is 96% of the power. For better comparison to the field the use of dB, i.e. this number in dB, as well as the other transmission and reflection data, would be helpful
- While 96% is a large number is it not 100%. The use of 'perfect' throughout the manuscript (including the title) is misleading and should be corrected.
- The work is motivated by 'energy conversion or information transfer can be maximized', a short description on how the presented MIPA could be find applications would help the reader.
- Fig 1 c and f would benefit from adding D and Q mode labels to the plot. CPA in the caption should be written in full form.
- How does the data from Fig 4 b and c compare to the data from Fig. 3? Under what angle is it taken (90 degree?) and is it new data or just a zoom-in?
- Absorption A depends on S_{11}^2 and S_{21}^2 , but most /all reflection and transmission data is plotted as S_{11} or S_{21} (not squared). The author might consider showing S_{11}^2 and S_{21}^2 throughout the manuscript, and use uniform axis notations (' S_{11} ' versus 'reflection' etc.)
- It is missing some technical information, such as a sketch of the measurement circuit.
- If ports 1 & 2 of the cavity are connected to the VNA, what are ports 3 & 4 (Fig. S2a) connected to? Presumably they are grounded via a 50 Ohm shunt, but if this is the case, it needs to be stated.
- What is the excitation level provided from the VNA for the measurements provided?
- Has the power dependence for the absorption been studied, and is the experiment done in the linear regime of FMR excitations
- Also, the length of the microstrip is given, but what is the width? Is it comparable to the 1 mm diameter of the YIG sphere?
- Moreover, the manuscript suffers from poor communication (e.g. the common acronym 'VNA' is used but never written out. Readers from outside the field will find this confusing).

There are sections that require heavy editing for clarity.

- For instance, the variables κ_d , κ_q , γ_d , and γ_q are not defined anywhere in the main text but are critical for understanding the theoretical underpinnings of the MIPA. While these are elucidated on in the supplementary information, this is not sufficient for readers to readily understand the concepts presented.
- The main text needs to include all variable definitions and explanations required to understand the central concepts.
- The chemical formula for YIG should also be provided.
- In addition to these issues, there are various grammar and phrasing issues throughout the text that can cause confusion.

It would be nice to see these questions being addressed. I do not recommend a publication at the current stage for these reasons.

Reviewer #2:

Remarks to the Author:

In this work, the authors experimentally investigate the perfect electromagnetic absorption in a cavity magnonics system. In previous studies, the perfect absorption was engineered via a single-port reflector or the destructive interference among multiple coherent incident beams. To obtain the complementary advantages of these two methods, the authors design a cavity magnonic system consisting of a cross cavity and a YIG sphere, where the Kittel mode in the YIG sphere is coupled to the dipole and quadrupole modes of the cavity. With the internal-mode structures, an input field can be perfectly absorbed by the hybrid system. Thus, the magnon-induced perfect absorption (MIPA) occurs.

The paper is quite interesting and well organized. The perfect absorption has been studied in various systems owing to its applications in, e.g., photon detector or stealth technology. Compared with existing methods for engineering perfect absorptions, the MIPA designed in this paper owns the simplicity and adjustability. According to the important results obtained in the manuscript, I would like to recommend the manuscript for publication in Nature Communications, provided the authors can address the following comments.

1. In a cavity magnon system, the coherent perfect absorption was demonstrated experimentally by Zhang et al (i.e., Ref. [57] in this manuscript). More discussions can tell the difference between the MIPA designed in this manuscript and the coherent perfect absorption implemented in Ref. [57].
2. In Ref. [57], the coherent perfect absorption can be used to engineer an effective gain of the cavity to produce a parity-time-symmetric Hamiltonian. I wonder whether the MIPA can be used to engineer an effective gain. Some discussions can be useful to readers.
3. In figures 2b and 2c, except the studied modes in this manuscript, there are some unexpected modes (i.e., small avoided crossings). Some brief discussions on their origins are also useful.

Reviewer #3:

Remarks to the Author:

Dear Editor,

In the submission "Interferometric control of magnon-induced perfect absorption in cavity magnonics," the authors demonstrated perfect absorption of a microwave system in a coupled cavity magnonic system. Coherent energy absorption in a physical system gains attention recently as the quanta of the energy carry information in a microscopic system for future electronic devices, e.g., spintronic and quantum devices. The authors employ magnonic elements, a sphere of yttrium iron garnet, and microwave resonators as the absorber, and four modes and four microwave signal ports contribute to the operation. They achieved nearly perfect absorption as high as 96 percent, which is impressive. The experimental demonstration, together with their comprehensive analysis of the complex physical system, is convincing. However, to warrant publication in Nature Communications, I would expect the authors to answer minor questions and revise their manuscript. If the authors convincingly explain these points in a revised manuscript, I support publication in Nature Communications.

Terminology and Typographical errors:

1. "so that the input signal can be completely trapped in the cavity", "so that all input power can be trapped in our device"
The term "trap" may give readers confusion in which the device can release energy to the incoming waveguide. I might suggest replacing it with the one having a meaning of dissipation.
2. "Their radio frequency (rf)-field distributions are calculated ..."
It would be helpful to indicate the color distribution shows the rf-magnetic field. Readers might have to refer to the supplementary information.
3. There seems to be a typographical error in the author's names of reference 59.
4. "The cross cavity used in our ..., which consists of two orthogonal half-wavelength microstrip line resonators with a length of 25 mm." in supplementary note 2.
The resonators in figure S2 seem co-planar waveguide ones. Microstrip line resonators do not have ground electrodes on the top surface of a substrate.
5. I suggest replacing H_{bat} in the supplementary note 3 with H_{bath} for better readability.
6. I suggest authors use "arg S₂₁ (deg)" instead of "P₂₁ (a.u)" in figure S6 in the supplementary note 6.

Comments:

7. Did the authors attach 50 Ohm loads to the unused microwave ports 3 and 4 to equalize the damping rates (κ_d and γ_d)?
8. Question: Where does the incoming energy dissipate?
It would be better to think where the incoming energy is consumed for future applications, such as microwave detectors, etc... The microwave signal may decay to the dipole, quadrupole, and magnetostatic modes through $\gamma_{d,x,y}$, γ_q , and γ_m . The energy might also dissipate to transmission lines 3 and 4 through the $\kappa_{d,y}$ of the dipole modes in the y-direction. Finding the fraction of each may give readers an intuitive understanding of magnon-induced perfect absorber.

Response Letter

Reply to comments of Reviewer #1

The manuscript presents a method for achieving magnon-induced perfect absorption (MIPA) using a cross cavity and a YIG sphere. The MIPA is achieved through tuning the interference between the hybridized modes which occur via coupling between the YIG magnon mode with the dipolar and quadrupolar photon modes of the cross cavity. The results are demonstrated well through the achievement of 96% power absorption, and the theoretical analysis of the hybridized system seems to model the data in a sufficient manner. Overall, the work presents a novel advancement in the field of cavity magnonics, and is theoretically sound. However, I have reservation about the conclusions drawn from their experimental observations, and would like to see more consistency within the data and more details on the data analysis before recommending publication in Nature Communications.

(Reply) We thank reviewer#1 for his/her appreciation as **“Overall, the work presents a novel advancement in the field of cavity magnonics, and is theoretically sound.”** We would like to thank the concerns and critical comments from reviewer#1, which allow us to improve the manuscript by addressing these issues. Below we respond to them one by one.

There are some issues with the manuscript:

- The maximum absorption is 96% of the power. For better comparison to the field the use of dB, i.e. this number in dB, as well as the other transmission and reflection data, would be helpful

(Reply) We thank the reviewer for this suggestion. We have squared S_{11} and S_{21} shown in Figs. 4(b) and (c) and converted them to dB scale, please see the Fig. R1 below. The absorption rate in Fig. R1 (b) is close to 0 dB.

Figure R1. (a) $|S_{11}|^2$, $|S_{21}|^2$ plotted in dB scale, which corresponds to the configuration of Fig. 4b in the main text. (b) Absorption rate of Fig.4c is replotted in dB scale, showing the value close to 0 dB.

According to the reviewer's suggestion, we have replaced Fig. 4(b) in the main text with the

dB scale. In Fig. 4(c) in the main text, we have added the scale dB numbers on the right side of this figure, while also retaining the previous linear scale on the left axis. The linear scale is suitable to exhibit the variations in absorption near 100%, and the labeled dB numbers can help the readers to make an intuitive comparison between these two different scales.

- While 96% is a large number is it not 100%. The use of 'perfect' throughout the manuscript (including the title) is misleading and should be corrected.

(Reply) We thank the reviewer for his/her question. Accordingly, we have modified the "perfect absorption" to "**nearly perfect absorption**" in our title and results part. We note that our method can achieve 100% absorption in theory. Our model takes advantage of the linear harmonic dynamics, so it's promising to be generalized to other harmonic systems or can be scaled down to the quantum regime with very few photons. We have added discussion as "**We note that, since our model...promising to be generalized to other harmonic systems, or scaled down to the quantum regime.**" in the right column on page 5.

- The work is motivated by 'energy conversion or information transfer can be maximized', a short description on how the presented MIPA could be find applications would help the reader.

(Reply) We thank the reviewer for this constructive comment. By further implementing the spin pumping effect in the magnetic material (*Physical Review Letters* 114, 227201 (2015)), the MIPA mechanism allows the incoming microwave to be mostly utilized for the generation and control of spin currents. Besides, by using an electrical detection technique, it can locally detect the spin current in an efficient way which stimulates ideas of magnon-based energy harvesting and sensing. In the main text, we have added some discussions as "**By implementing the...for the magnon-based energy harvesting and sensing**" in the right column on page 5 in the conclusion part.

- Fig 1 c and f would benefit from adding D and Q mode labels to the plot. CPA in the caption should be written in full form.

(Reply) We thank the reviewer for his/her advice. We have added D and Q labels to the plot in Fig. 1, and added "**...D- and Q-modes (labelled near their resonances).**" in the figure caption. Besides, we rewrite CPA in the full form of "**coherent perfect absorption (CPA)**" in the caption.

- How does the data from Fig 4 b and c compare to the data from Fig. 3? Under what angle is it taken (90 degree?) and is it new data or just a zoom-in?

(Reply) The experimental results plotted in Fig.4 were taken at 90 degree, and we have added "**At the condition of $\theta = 90^\circ$, as the upper-CMP mode.....**" in the right column on page 5 to clarify this issue. These results were measured from a modified cross cavity with optimized ports, as we stated in the main text as "**Accordingly, we make another cross cavity, keeping...**" in the right column on page 5.

- Absorption A depends on S_{11}^2 and S_{21}^2 , but most /all reflection and transmission data

is plotted as S11 or S21 (not squared). The author might consider showing S_{11}^2 and S_{21}^2 throughout the manuscript, and use uniform axis notations ('S11' versus 'reflection' etc.)

(Reply) Thank the reviewer for his/her suggestion. Accordingly, we have replaced $|S_{11}|$, $|S_{21}|$ in Fig.3 (a) (b) and (e) (f) with $|S_{11}|^2$, $|S_{21}|^2$. In Fig. 4(b), the $|S_{11}|$ and $|S_{21}|$ have also been squared and changed to a dB scale as suggested in question 1.

- It is missing some technical information, such as a sketch of the measurement circuit.

(Reply) We thank the reviewer for this helpful comment. In Fig.1(d), we've added a sketch of the measurement circuit.

- If ports 1 & 2 of the cavity are connected to the VNA, what are ports 3 & 4 (Fig. S2a) connected to? Presumably they are grounded via a 50 Ohm shunt, but if this is the case, it needs to be stated.

(Reply) We appreciate the reviewer for pointing out this issue. Because the quadrupole (Q) mode simultaneously couples to four ports in such a symmetric device, there exists a weak signal leakage from ports 3 & 4 if they are connected to the matched microwave circuit. In experiment, this issue was overcome by using short terminators, which produced a reflection at ports 3 & 4 and prevented the energy loss from these two ports. The only purpose for using ports 3 & 4 is to characterize the cavity modes in our cross cavity, especially the D-mode along the y-axis.

To avoid misleading readers, we add "..... from port-1 and port-2 are shown in Figs. 1 (e) and (f) with two other ports shorted." in the left column on page 2, as well as add a paragraph at the end of the supplementary Note 2: "**Ports 3 and 4 of the cross cavity were used to characterize two distinct cavity modes (D-modes and Q-mode), especially the perpendicular D-mode along the y-axis. When we measured the microwave absorption, ports 3 and 4 of our device were blocked by two short terminators to prevent weak signal leakages.**"

- What is the excitation level provided from the VNA for the measurements provided?

(Reply) We thank the reviewer for pointing out our omission. The excitation level of VNA is -5 dBm. Correspondingly, we have added a sentence in the left column on the page 2 as "...measured by Vector Network Analyzer (VNA) with an excitation power of -5 dBm from port-1 to port-2.....".

- Has the power dependence for the absorption been studied, and is the experiment done in the linear regime of FMR excitations

(Reply) We thank the reviewer for his/her question. During our experiment, the excitation power of the VNA is set to a low level to ensure the excitation of FMR in the linear regime. The estimated intracavity photon number in our system is about 10^{11} , and the estimated spin number in the 1-mm diameter YIG sphere is about 10^{19} . The number of photons is orders of magnitude smaller than the number of spins. Therefore, we may infer that the magnon dynamics stays in the linear regime. The measured reflection, transmission and absorption are independent of input power.

- Also, the length of the microstrip is given, but what is the width? Is it comparable to the 1 mm diameter of the YIG sphere?

(Reply) We thank the reviewer for his/her nice question. The width of the central conductor is

1.14 mm, which is comparable to the diameter of the YIG sphere. Accordingly, we add the width of the central conductor in the Method section ".....and a central conductor width of 1.14 mm. "

- Moreover, the manuscript suffers from poor communication (e.g. the common acronym 'VNA' is used but never written out. Readers from outside the field will find this confusing). There are sections that require heavy editing for clarity.

(Reply) We thank the reviewer for his/her suggestion. We have changed the VNA as "**Vector Network Analyzer (VNA)**" in the left column on page 2 in our manuscript.

- For instance, the variables κ_d , κ_q , γ_d , and γ_q are not defined anywhere in the main text but are critical for understanding the theoretical underpinnings of the MIPA. While these are elucidated on in the supplementary information, this is not sufficient for readers to readily understand the concepts presented.

(Reply) We thank the reviewer very much for pointing out this issue. κ_d , κ_q , γ_d and γ_q respectively represent the external and intrinsic damping rates of the D-modes and the Q-mode. According to reviewer's suggestion, we have added an explanation after Eq. (2) in the left column on page 3 as " **$\kappa_{d,q}$ and $\gamma_{d,q}$ respectively represent the external and intrinsic damping rates of the D-modes and the Q-mode. γ_m is the intrinsic damping rate of the magnon mode**".

- The main text needs to include all variable definitions and explanations required to understand the central concepts.

(Reply) Thanks very much for reviewer's suggestion. We have carefully checked our manuscript and given all variables clear definitions. For example, (i) the CMP term in the abstract has now been revised as "**cavity magnon-polariton**". (ii) We define the angle θ in a clearer way, as "The bias magnetic field H is applied in the x-y plane **with an adjustable angle θ from the y-axis.**" in the caption of Fig.1 and "A bias magnetic field H with a tunable angle (**θ from the y-axis**) shown in" in the right column on page 2. (iii) We add the definition of γ_+ in the right column on page 4, as "....., **where γ_+ is the intrinsic damping rate of the upper-CMP mode**".

- The chemical formula for YIG should also be provided.

(Reply) We thank the reviewer for his/her suggestion. Now, the chemical formula of YIG is provided in the main text as "**...an yttrium iron garnet (YIG, $Y_3Fe_5O_{12}$) sphere...**" in the left column on page 2.

- In addition to these issues, there are various grammar and phrasing issues throughout the text that can cause confusion.

(Reply) We thank the reviewer for correcting us. In this version, we have improved the readability of our manuscript by fixing the grammar and phrasing problems.

It would be nice to see these questions being addressed. I do not recommend a publication at the current stage for these reasons.

Reply to comments of Reviewer #2

In this work, the authors experimentally investigate the perfect electromagnetic absorption in a cavity magnonics system. In previous studies, the perfect absorption was engineered via a single-port reflector or the destructive interference among multiple coherent incident beams. To obtain the complementary advantages of these two methods, the authors design a cavity magnonic system consisting of a cross cavity and a YIG sphere, where the Kittel mode in the YIG sphere is coupled to the dipole and quadrupole modes of the cavity. With the internal-mode structures, an input field can be perfectly absorbed by the hybrid system. Thus, the magnon-induced perfect absorption (MIPA) occurs.

The paper is quite interesting and well organized. The perfect absorption has been studied in various systems owing to its applications in, e.g., photon detector or stealth technology. Compared with existing methods for engineering perfect absorptions, the MIPA designed in this paper owns the simplicity and adjustability. According to the important results obtained in the manuscript, I would like to recommend the manuscript for publication in Nature Communications, provided the authors can address the following comments.

(Reply) We thank reviewer#2 for his/her high recognition as **“the MIPA designed in this paper owns the simplicity and adjustability”** and **“I would like to recommend the manuscript for publication in Nature Communications...”**. We found the comments from reviewer#2 to be very helpful. Following these comments, we have answered the reviewer’s concerns and improved the manuscript, with details shown in the following parts.

1. In a cavity magnon system, the coherent perfect absorption was demonstrated experimentally by Zhang et al (i.e., Ref. [57] in this manuscript). More discussions can tell the difference between the MIPA designed in this manuscript and the coherent perfect absorption implemented in Ref. [57].

(Reply) We thank the reviewer for this question. Zhang’s work firstly demonstrated the coherent perfect absorption (CPA) in cavity magnonics. It was realized by a two-tone experiment, where two coherent input signals provided effective gains to the coupled cavity photon-magnon system. In this manner, interesting phenomena, including the exceptional point, PT symmetry and pseudo-Hermiticity, were observed.

The successful demonstration of two-tone experiment by Zhang et al. motivates us to find solution for single-beam perfect absorption in cavity magnonics. In our work, we develop a new method to achieve perfect absorption, i.e., magnon-induced perfect absorption (MIPA), by using the magnon-mediated interference between two resonant channels. This method enables the realization of single-beam perfect absorption in one-tone measurements on cavity magnonic devices without breaking the devices’ boundary symmetry, which may open a new route for single-beam perfect absorption applications.

In the main text, we have added the discussion to compare our work with Ref. [57] (now Ref. [35] in the revised manuscript), as **“In particular, with two input beams as effective gains, CPA has been realized in a coupled magnon-photon system. This motivates us to further explore solutions for single-beam perfect absorption in cavity magnonics.”**

2. In Ref. [57], the coherent perfect absorption can be used to engineer an effective gain of the cavity to produce a parity-time-symmetric Hamiltonian. I wonder whether the MIPA can be used to engineer an effective gain. Some discussions can be useful to readers.

(Reply) We thank the reviewer for this inspiring question. According to the Ref. [57] (now Ref. [35] in the revised manuscript), the extrinsic damping rates of the upper-branch polariton mode and the Q-mode, i.e., κ_+ and κ_q , may also be used as effective gains to construct a parity-time-symmetric system. In the main text, we have added some discussions in the conclusion part on page 5 as **“The extrinsic damping rates enriched by pseudo-Hermiticity and exceptional-point physics”**.

3. In figures 2b and 2c, except the studied modes in this manuscript, there are some unexpected modes (i.e., small avoided crossings). Some brief discussions on their origins are also useful.

(Reply) Besides the Kittel mode at the frequency of ω_m , two other non-uniform magnon modes are observed. Two small avoided crossings in Figs. 2(b) and 2(c) arise from the magnetic dipole interactions between non-uniform magnon modes and cavity modes. Their mode frequencies are far-detuned from the Kittel mode ω_m , so that they have negligible influences on the performance of MIPA.

Reply to comments of Reviewer #3

Dear Editor,

In the submission “Interferometric control of magnon-induced perfect absorption in cavity magnonics,” the authors demonstrated perfect absorption of a microwave system in a coupled cavity magnonic system. Coherent energy absorption in a physical system gains attention recently as the quanta of the energy carry information in a microscopic system for future electronic devices, e.g., spintronic and quantum devices. The authors employ magnonic elements, a sphere of yttrium iron garnet, and microwave resonators as the absorber, and four modes and four microwave signal ports contribute to the operation. They achieved nearly perfect absorption as high as 96 percent, which is impressive. The experimental demonstration, together with their comprehensive analysis of the complex physical system, is convincing. However, to warrant publication in Nature Communications, I would expect the authors to answer minor questions and revise their manuscript. If the authors convincingly explain these points in a revised manuscript, I support publication in Nature Communications.

(Reply) We thank the reviewer #3 for the assessment as **“They achieved nearly perfect**

absorption as high as 96 percent, which is impressive. The experimental demonstration, together with their comprehensive analysis of the complex physical system, is convincing.”

We appreciate the reviewer’s constructive suggestions, which are so helpful for us to improve the manuscript. Please see below for our point-to-point response.

Terminology and Typographical errors:

1. "so that the input signal can be completely trapped in the cavity", "so that all input power can be trapped in our device"

The term “trap” may give readers confusion in which the device can release energy to the incoming waveguide. I might suggest replacing it with the one having a meaning of dissipation.

(Reply) We agree with the reviewer. Accordingly, we have revised these two sentences as **“so that the input signal can be completely dissipated in the cavity”** and **“so that all input power can be absorbed and then dissipated in our device.”** in the main text.

2. “Their radio frequency (rf-)field distributions are calculated ...”

It would be helpful to indicate the color distribution shows the rf-magnetic field. Readers might have to refer to the supplementary information.

(Reply) We thank the reviewer for this comment. We have added a color scale below the color distribution in Fig. 1.

3. There seems to be a typographical error in the author's names of reference 59.

(Reply) We thank the reviewer for his/her careful reading, we have corrected this error in Ref. [59], as **“Lachance-Quirion, D., Tabuchi, Y., Gloppe, A., Usami, K. & Nakamura, Y. Hybrid quantum systems based on magnonics. Applied Physics Express 12, 070101 (2019).”**

”

4. “The cross cavity used in our ..., which consists of two orthogonal half-wavelength microstrip line resonators with a length of 25 mm.” in supplementary note 2.

The resonators in figure S2 seem co-planar waveguide ones. Microstrip line resonators do not have ground electrodes on the top surface of a substrate.

(Reply) We thank the reviewer for pointing out our mistake. We have changed the term “microstrip line resonator” to **“coplanar waveguide resonator”** in our manuscript in the left column on page 2, as well as in the supplementary note on page 2.

5. I suggest replacing H_{bat} in the supplementary note 3 with H_{bath} for better readability.

6. I suggest authors use “arg S21 (deg)” instead of “P21 (a.u)” in figure S6 in the supplementary note 6.

(Reply to Q5 & 6) We thank the reviewer for these two suggestions. Accordingly, we have corrected them in the supplementary note in equation (S6) and Fig. S6, respectively.

7. Did the authors attach 50 Ohm loads to the unused microwave ports 3 and 4 to equalize the damping rates (κ_d and γ_d)?

(Reply) We thank the reviewer for his/her thoughtful question. γ_d is the intrinsic damping rate of two D-modes, which is mainly determined by the dielectric loss of the substrate

material and the conductance loss of the metal surface. The loading status of ports 3 and 4 has an ignorable influence on γ_d . The term κ_d is the external damping rate of two D-modes, which is mainly determined by the coupling gaps between the port and the end of the cross cavity. In experiment, we connected ports 3 and 4 with two short terminators to prevent weak signal leakages. The loading status of ports 3 and 4 may slightly perturbate the external damping of the perpendicular D-mode (along the y-axis). However, from our measured spectra, the external damping rates of two D-modes were almost equal, so that we use the same κ_d to represent them.

8. Question: Where does the incoming energy dissipate?

It would be better to think where the incoming energy is consumed for future applications, such as microwave detectors, etc... The microwave signal may decay to the dipole, quadrupole, and magnetostatic modes through $\gamma_{d,x,y}$, γ_q , and γ_m . The energy might also dissipate to transmission lines 3 and 4 through the $\kappa_{d,y}$ of the dipole modes in the y-direction. Finding the fraction of each may give readers an intuitive understanding of magnon-induced perfect absorber.

(Reply) We thank the reviewer for his/her thoughtful question. In our current experimental set-up, there is no energy harvesting element, so that all the absorbed microwave energy will be eventually dissipated into heat. During this process, there exist several possible dissipation channels:

1. If without connecting terminators, there may exist small energy leakage from ports 3 and 4. In experiment, to avoid it, ports 3 and 4 were connected to two short terminators.
2. Without the magnon mode, the energy absorption rate of the bare Q-mode is about 50% (as shown by the black solid line in Fig. 4 (c)). This energy will be eventually dissipated into heat through the γ_q .
3. When the upper-branch polariton mode (ω_+) approaches the Q-mode, nearly MIPA with an absorption rate of 96% is achieved. With respect to the bare Q-mode, the increment of the absorption rate is from the upper-branch polariton mode (ω_+), which is about 46%. This part of energy will be eventually dissipated into heat through the γ_+ which is a function of γ_d and γ_m .

For new applications enabled by our method, one of the immediate capabilities is to replace the currently used YIG sphere with a spintronic device, such as a spin pumping device (*Physical Review Letters* 114, 227201 (2015)), so that microwave energy can be efficiently converted into a spin current via the MIPA. And this spin current can be further electrically detected and used for information processing.

Reviewers' Comments:

Reviewer #1:

None

Reviewer #2:

Remarks to the Author:

The authors have satisfactorily answered my questions and also improved the manuscript following my comments. Therefore, I recommend this revised manuscript for publication in Nature Communications.

Reviewer #3:

Remarks to the Author:

Dear Editor,

In the resubmission of a manuscript, the authors addressed raised concerns and answered my questions. The authors sincerely explained the effect of the loss mechanism in the response letter, and it is convincing. The nearly perfect absorption of 96%, as well as their comprehensive analysis of tricky resonant structure including three microwave modes and the Kittel mode, is impressive. Thus, I support publication in Nature Communication.

Response Letter

Reply to comments of Reviewer #2

The authors have satisfactorily answered my questions and also improved the manuscript following my comments. Therefore, I recommend this revised manuscript for publication in Nature Communications.

(Reply) We greatly thank the reviewer for his/her positive evaluation of our work and the recommendation for publication.

Reply to comments of Reviewer #3

In the resubmission of a manuscript, the authors addressed raised concerns and answered my questions. The authors sincerely explained the effect of the loss mechanism in the response letter, and it is convincing. The nearly perfect absorption of 96%, as well as their comprehensive analysis of tricky resonant structure including three microwave modes and the Kittel mode, is impressive. Thus, I support publication in Nature Communication.

(Reply) We appreciate the positive comments of the reviewer as “the effect of the loss mechanism is convincing” and “their comprehensive analysis is impressive”. We thank the reviewer for recommending our article for publication.